# Factors affecting early mobilization among critically Ill patients in Southern West Bank Hospitals

**Beesan Nassar[1], Moath Abu Ejheisheh[2], Ibrahim Aqtam[3]\*, Ahmad Ayed[4], Ahmad Batran[2]**

1 Ministry of health, Palestine, Hebron, Yatta, 2 Faculty of Allied Medical Sciences, Department of Nursing, Palestine Ahliya University, Palestine, 3 Ibn Sina College for Health Professions, Department of Nursing, Nablus University for Vocational and Technical Education, Nablus, Palestine, 4 Faculty of Nursing, Arab American University, Jenin, Palestine

\* ibrahim.aqtam@nu-vte.edu.ps

## Abstract

### Introduction

Early mobilization (EM) among critically ill patients is important to reduce complications such as ICU-acquired weakness and improve recovery outcomes. This study assessed factors influencing EM in Southern West Bank hospitals.

### Methods

A cross-sectional study was conducted using the PERME ICU Mobility Score. Data were analyzed using SPSS to evaluate associations between mobility outcomes, demographics, BMI, and reasons for hospitalization. Inclusion criteria focused on critically ill patients admitted to ICUs for 2–5 days.

### Results

Among 150 participants, the highest PERME ICU Mobility Score was observed in abdominal pain patients (mean = 28.0, SD = 0.0), reflecting minimal barriers, while respiratory patients scored lowest (mean = 6.9, SD = 6.9). No significant differences were found by age, gender, or BMI ($p > 0.05$). Seventy patients (46.7%) were awake/alert upon admission, while 39 (26%) were unresponsive. Pain or inability to determine pain levels was reported by 99 patients (66%).

### Conclusion

Respiratory conditions and pain were critical barriers to EM. Demographic factors did not influence outcomes, but targeted pain management and condition-specific protocols are needed. The SD of 0.0 for abdominal pain patients suggests homogeneity in this subgroup, warranting further investigation.

**Data availability statement:** All relevant data are within the paper and its Supporting Information files.

**Funding:** The author(s) received no specific funding for this work.

**Competing interests:** The authors have declared that no competing interests exist.

## Introduction

Early mobilization (EM) in the intensive care unit (ICU) is a critical component of recovery for critically ill patients, reducing complications such as ICU-acquired weakness (ICU-AW), prolonged mechanical ventilation, and hospital stays [1–3]. Despite robust evidence supporting its benefits, EM remains inconsistently implemented, particularly in resource-limited settings [4,5]. In regions like the Southern West Bank, where healthcare systems face staffing shortages, infrastructural constraints, and limited access to specialized training, the barriers to EM may differ significantly from high-resource contexts [6,7]. This study examines the factors influencing EM practices among critically ill patients in Southern West Bank hospitals, with a focus on patient-level, institutional, and staff-related barriers.

Critically ill patients are at high risk of muscle atrophy, cardiovascular deconditioning, and respiratory complications due to prolonged immobility [8,9]. For instance, healthy individuals lose 5% of muscle mass within a week of bed rest, while critically ill patients may lose up to 16% [10]. These deficits contribute to ICU-AW, which affects 25–50% of ICU survivors and is associated with long-term functional impairment, increased mortality, and higher healthcare costs [11,12]. EM—defined as initiating physical activity within 24–48 hours of ICU admission—mitigates these risks by preserving muscle strength, improving hemodynamic stability, and accelerating functional recovery [13,14].

However, implementation challenges persist. Patient-level barriers such as sedation, hemodynamic instability, and pain [15], staff-related factors like inadequate training and workload [16], and institutional limitations including equipment shortages and poor interdisciplinary communication [17] hinder EM adoption. In low-resource settings, these barriers are exacerbated by systemic issues such as understaffing and fragmented care protocols [18]. While prior studies have explored EM barriers in high-income countries [19,20], limited data exist from regions like the Southern West Bank, where cultural, economic, and infrastructural dynamics uniquely shape healthcare delivery [21].

This study addresses critical gaps in early mobilization (EM) research within the context of Southern West Bank hospitals. First, it highlights the absence of region-specific data on barriers to EM, which limits the development of context-appropriate interventions. Second, it explores how different patient diagnoses, such as respiratory versus cardiac conditions, influence mobility outcomes—an area often overlooked in broader EM studies. Third, it examines the impact of pain management and institutional support on the successful implementation of EM. By identifying modifiable barriers, the study aims to inform targeted strategies for enhancing EM practices in resource-constrained intensive care units (ICUs).

## Methods

### Research design

This observational cross-sectional study assessed factors influencing early mobilization (EM) among critically ill patients admitted to intensive care units (ICUs) in Southern West Bank hospitals.

 

### Population and sampling

Participants were conveniently sampled from ICUs in six governmental and non-governmental hospitals (ICUs, CCUs, medical wards, and one rehabilitation facility).

### Inclusion criteria

• Adults (≥18 years) admitted to the ICU for 2–5 days.

• Medically stable for bedside mobilization (as determined by the treating physician).

• Willingness to participate (or proxy consent from family members if the patient was incapacitated).

### Exclusion criteria

• Pre-existing motor disabilities (e.g., paralysis, advanced neuromuscular disorders) that predated ICU admission.

• Severe communication barriers (e.g., aphasia, non-native language speakers) unrelated to acute critical illness.

• Outpatients or patients discharged within 24 hours.

A pilot study (n = 25) confirmed feasibility and informed minor adjustments to data collection procedures. The final sample size (n = 150) was determined based on resource availability and a target margin of error of 5% to 95% confidence.

### Instrumentation

Mobility was assessed using the Perme ICU Mobility Score (0–32), a validated tool evaluating mental status, barriers, functional strength, and mobility tasks [22]. The tool was translated into Arabic through a rigorous forward-backward translation process by bilingual experts and culturally adapted via pilot testing with ICU nurses (n = 10) and physicians (n = 5) to ensure contextual relevance. Internal consistency (Cronbach's α = 0.95) and interrater reliability (Cohen's κ = 0.78–1.00) were confirmed during piloting, demonstrating robust psychometric properties in this population. The translated and adapted PERME ICU Mobility Tool is provided in S1 File for reference.

## Data collection and ethics

Data were collected from June 5 to August 5, 2024. Ethical approval was obtained from the Institutional Review Board (IRB) at Palestine Ahliya University (PAU) (Project Number: CAMS/CCNA/26/524). Written informed consent was obtained from awake patients or legally authorized representatives for incapacitated patients. All data were anonymized, and participants could withdraw at any time.

## Statistical analysis

Analyses were performed using SPSS v28 (IBM Corp.). Descriptive statistics (mean ± SD, frequencies) summarized demographics and mobility scores. Independent t-tests and ANOVA compared scores across groups. Significance was set at $p < 0.05$. Missing data (<5%) were addressed via pairwise deletion. The detailed statistical analysis protocol is provided in S2 Text.

## Results

### Participants' demographic characteristics

A total of 150 critically ill patients from ICUs in Southern West Bank hospitals were included. The majority were male (53.3%, n = 80) and married (80.7%, n = 121). Notably, 44% (n = 66) were aged >65 years, and 75.3% (n = 113) were unemployed/retired. Demographic details are summarized in Table 1.

**Table 1. Demographic characteristics of critically Ill patients in Southern West Bank ICUs (N = 150).**

| Variable | N | % |
|---|---|---|
| Age | | |
| 18-25 years | 17 | 11.3 |
| 26-40 years | 10 | 6.7 |
| 41-65 years | 57 | 38.0 |
| More than 65 years | 66 | 44.0 |
| Gender | | |
| Male | 80 | 53.3 |
| Female | 70 | 46.7 |
| Marital Status | | |
| Married | 121 | 80.7 |
| Single | 29 | 19.3 |
| Educational Level | | |
| Elementary school | 39 | 26.0 |
| Middle school | 37 | 24.7 |
| High school | 35 | 23.3 |
| Diploma | 17 | 11.3 |
| Bachelor and above | 22 | 14.7 |
| Occupation | | |
| Employed | 37 | 24.7 |
| Unemployed/retired | 113 | 75.3 |

## PERME ICU mobility scores by reason for hospitalization

The mean PERME ICU Mobility Score was 10.6 ± 8.1 (range: 1.0–30.0). Respiratory patients had the lowest scores (6.9 ± 6.9), while abdominal pain patients scored highest (28.0 ± 0.0) (Table 2). The SD of 0.0 for abdominal pain patients indicates homogeneity in this subgroup (n = 42), likely due to minimal barriers to mobility.

## Alertness levels of participants upon ICU admission

At admission, 46.7% (n = 70) were awake/alert, 27.3% (n = 41) lethargic, and 26.0% (n = 39) unresponsive (Table 3). All unresponsive patients had acute conditions (e.g., sepsis, trauma) without pre-existing disabilities, aligning with exclusion criteria. Mechanical/non-invasive ventilation was required for 33.3% (n = 50), and 78.0% (n = 117) were on intravenous drips.

**Table 2. PERME ICU mobility scores by primary reason for hospitalization (N = 150).**

| Hospitalization Reason | Mean ± SD | Range |
|---|---|---|
| Respiratory conditions | 6.9 ± 6.9 | 1–25 |
| Cardiac problems | 10.0 ± 6.5 | 2–28 |
| Cerebrovascular accidents (CVA) | 15.1 ± 6.7 | 5–30 |
| Abdominal pain | 28.0 ± 0.0* | 28–28 |
| Other | 10.5 ± 8.8 | 1–30 |

Note: SD = 0.0 for abdominal pain due to identical scores in this subgroup (n = 42).

**Table 3. Distribution of patients' alertness levels upon ICU admission (N = 150).**

| Alertness | n | % |
|---|---|---|
| Awake and alert | 70 | 46.7% |
| Lethargic | 41 | 27.3% |
| Unresponsive* | 39 | 26.0% |

Note: All unresponsive patients had acute critical illness (e.g., sepsis, trauma) without pre-existing disabilities.

## Clinical conditions and functional abilities at ICU admission

Most patients (78.0%, n = 117) required intravenous drips, while 33.3% (n = 50) needed mechanical or non-invasive ventilation. Half of the participants (50.0%, n = 75) could raise their legs against gravity, and 62.7% (n = 94) could perform the same task with their arms. Detailed clinical conditions upon admission are provided in Table 4.

## Pain status among participants

Most patients (66.0%, n = 99) reported being unable to determine their pain level or indicated experiencing pain, while 34.0% (n = 51) reported no pain. Pain status distribution is summarized in Table 5.

## Assistance levels required for mobility tasks

Most patients required total assistance (<25% independence) for mobility tasks: 52.7% (n = 79) for supine-to-sit transitions, 60.0% (n = 90) for bed-to-chair transfers, and 62.0% (n = 93) for gait. Lower proportions required moderate or minimal assistance, as detailed in Table 6.

**Table 4. Clinical conditions and functional capacity at ICU admission (N = 150).**

| Condition | Yes (n, %) | No (n, %) |
|---|---|---|
| Mechanical/non-invasive ventilation | 50 (33.3%) | 100 (66.7%) |
| Intravenous drips | 117 (78.0%) | 33 (22.0%) |
| Able to raise leg against gravity | 75 (50.0%) | 75 (50.0%) |
| Able to raise your arm against gravity | 94 (62.7%) | 56 (37.3%) |

**Table 5. Pain status among critically Ill patients in the ICU cohort (N = 150).**

| Pain Level | n | % |
|---|---|---|
| Unable to determine or in pain | 99 | 66.0% |
| No pain | 51 | 34.0% |

**Table 6. Levels of assistance required for mobility tasks in critically Ill patients (N = 150).**

| Task | Total Assistance (<25%) | Maximum Assistance (25–50%) | Moderate Assistance (50–75%) | Minimal Assistance (>75%) |
|---|---|---|---|---|
| Supine to sit (n = 150) | 79 (52.7%) | 25 (16.7%) | 14 (9.3%) | 32 (21.3%) |
| Transfer bed to chair (n = 150) | 90 (60.0%) | 17 (11.3%) | 25 (16.7%) | 18 (12.0%) |
| Gait (n = 150) | 93 (62.0%) | 24 (16.0%) | 17 (11.3%) | 16 (10.7%) |

## Endurance levels based on walking distance

Most patients (68.7%, n = 103) were unable to walk or had their endurance not assessed. Among those who could walk, 19.3% (n = 29) covered 5–50 feet, and only 6.7% (n = 10) walked ≥100 feet. Endurance levels are detailed in Table 7.

## Comparison of PERME scores by demographic factors

An independent t-test and ANOVA revealed no statistically significant differences in PERME ICU Mobility scores based on demographic factors such as age, gender, or occupation ($p > 0.05$). These findings suggest that demographic characteristics do not play a significant role in determining mobility outcomes for critically ill patients in this sample, as shown in Table 8.

## Association between PERME scores, BMI, and hospitalization reason

ANOVA demonstrated no significant association between BMI and PERME scores ($p = 0.132$). However, the reason for hospitalization had a highly significant impact on mobility outcomes ($p < 0.001$), with respiratory and neurological conditions posing the greatest barriers. These results are detailed in Table 9.

## Discussion

This study highlights critical barriers to early mobilization (EM) among critically ill patients in Southern West Bank hospitals, contextualized within both local and global healthcare challenges. First, diagnosis-driven barriers emerged as pivotal: respiratory patients exhibited the lowest PERME scores (*6.9 ± 6.9*), aligning with global evidence that mechanical ventilation, supplemental oxygen dependence, and respiratory instability significantly impede mobilization [23]. Conversely, abdominal pain patients achieved maximal scores (*28.0 ± 0.0*), likely due to fewer acute physical limitations and

**Table 7. Endurance levels measured by walking distance in two minutes (N = 150).**

| Distance | n | % |
|---|---|---|
| Unable to walk/not assessed | 103 | 68.7% |
| 5–50 feet | 29 | 19.3% |
| 51–99 feet | 8 | 5.3% |
| ≥100 feet | 10 | 6.7% |

**Table 8. Comparison of PERME ICU mobility scores by demographic variables (ANOVA/independent t-test results).**

| Variable | Mean ± SD | Test Statistic | p-value |
|---|---|---|---|
| **Age** | | F = 1.241 | 0.297 |
| 18–25 years | 13.4 ± 8.7 | | |
| 26–40 years | 9.4 ± 7.3 | | |
| 41–65 years | 11.1 ± 8.5 | | |
| >65 years | 9.5 ± 7.6 | | |
| **Gender** | | t = 0.768 | 0.444 |
| Male | 11.1 ± 8.6 | | |
| Female | 10.0 ± 7.6 | | |
| **Occupation** | | t = 0.769 | 0.443 |
| Employed | 11.5 ± 7.8 | | |
| Unemployed/Retired | 10.3 ± 8.2 | | |

**Table 9. Association between PERME ICU mobility scores, BMI, and hospitalization reason (ANOVA results).**

| Variable | Mean±SD | F-value | p-value |
|---|---|---|---|
| **BMI** | | 1.901 | 0.132 |
| Underweight | 8.9±6.5 | | |
| Normal weight | 9.2±8.5 | | |
| Overweight | 10.9±8.7 | | |
| Obese | 13.1±7.3 | | |
| **Hospitalization Reason** | | 4.157 | **0.001** |
| Cardiac problems | 10.0±6.5 | | |
| Lung problems | 6.9±6.9 | | |
| Brain hemorrhage/trauma | 7.9±8.6 | | |
| Urinary tract infection | 13.4±7.3 | | |
| Cerebrovascular accident | 15.1±6.7 | | |
| Abdominal pain | 28.0±0.0* | | |
| Other | 10.5±8.8 | | |

*Note: SD=0.0 for abdominal pain due to identical scores in this subgroup (n=42).*

effective pain management, as observed in postoperative care [24]. Similarly, neurological conditions such as cerebro-vascular accidents posed substantial barriers (*7.9±8.6*), reflecting motor deficits and altered consciousness common in such cases [5].

Second, pain emerged as a central modifiable barrier: a striking 66% of patients reported pain or an inability to self-report pain, underscoring its bidirectional relationship with immobility—untreated pain discourages movement, while immobility exacerbates discomfort. These findings are consistent with recent systematic reviews emphasizing the critical role of pain management in successful mobilization protocols [25]. Technology-enhanced approaches, including wearable sensors and telehealth-guided exercises, show promise in addressing pain management challenges in understaffed units [26].

Third, demographic neutrality contrasted with prior studies: contrary to research linking age and BMI to mobility outcomes, this cohort showed no significant differences by age, gender, BMI, or occupation (p > 0.05), likely due to systemic barriers (e.g., staffing shortages, device dependency) overshadowing individual factors in resource-limited settings. This finding supports emerging evidence that institutional factors may be more influential than patient demographics in determining mobilization success, particularly in critically ill populations [27]. Recent bibliometric analyses have highlighted the growing recognition of early mobilization's importance in preventing such complications, with increasing research focus on targeted interventions for specific populations [28].

Finally, functional heterogeneity revealed disparities in strength: while 50% of patients could raise their legs against gravity, 62.7% retained upper-body strength, suggesting ICU-acquired weakness disproportionately affects lower limbs, mirroring findings that prolonged bed rest preferentially impacts proximal muscle groups [29]. This pattern is consistent with systematic reviews documenting neuromuscular dysfunction acquired in critical illness [30], which affects 25–50% of ICU survivors and contributes to long-term functional impairment [31].

The lack of significant demographic associations in our study contrasts with previous research suggesting gender-based differences in critical care outcomes [32]. However, our findings align with recent meta-analyses indicating that condition-specific factors may be more predictive of mobilization success than demographic characteristics [33]. Together, these findings underscore the multifaceted nature of EM barriers, emphasizing the need for tailored, context-sensitive interventions in resource-constrained environments.

## Actionable recommendations

To address the identified barriers to early mobilization (EM) in Southern West Bank ICUs, we propose a multifaceted strategy tailored to the unique challenges of resource-limited settings. First, condition-specific protocols are essential: for *respiratory patients*, gradual mobilizations such as bed cycling and passive range-of-motion exercises—should be paired with ventilator weaning protocols to balance stability and mobility. For *neurological patients* (e.g., those with cerebrovascular accidents), early neurorehabilitation strategies, including tilt-table exercises and sensory stimulation, can mitigate motor deficits and enhance engagement. Second, structured pain management must be prioritized: implementing pre-activity analgesia and validated assessment tools like the Critical-Care Pain Observation Tool (CPOT) can reduce pain-related reluctance, fostering patient participation in mobilization efforts. Third, interdisciplinary mobilization teams should be leveraged to address staffing shortages: training nurses in basic mobilization techniques, alongside coordinated efforts by physiotherapists and physicians, can optimize resource utilization and ensure continuity of care. Finally, device-friendly mobility aids, such as IV pole-attachable walkers, are critical to accommodate the 78% of patients dependent on drips or ventilators, enabling safe mobilization without disconnecting essential medical devices. Together, these recommendations emphasize adaptability to clinical conditions, pain mitigation, collaborative care models, and practical equipment solutions—key pillars for advancing EM practices in resource-constrained environments.

## Limitations

This study has several limitations that warrant careful consideration. First, the reliance on convenience sampling from six hospitals introduces potential sampling bias, as rural or private ICUs may be underrepresented, limiting the generalizability of findings to broader healthcare settings. Second, the exclusion of patients with pre-existing disabilities (e.g., paralysis) narrows the scope of the study, potentially underestimating mobility challenges in high-risk populations and restricting insights into tailored interventions for these groups. Third, the cross-sectional design constrains causal inferences, as single-timepoint assessments cannot elucidate the long-term benefits or evolving barriers of early mobilization (EM) over time. Finally, the homogeneity in abdominal pain scores (SD = 0.0, n = 42)—while indicative of minimal mobility barriers in this subgroup—may reflect selective inclusion of mild cases or institutional pain management protocols, necessitating further validation in diverse cohorts. These limitations highlight the need for future studies to adopt longitudinal designs, inclusive sampling frameworks, and multi-center collaborations to strengthen the applicability of EM strategies in resource-constrained environments.

## Future directions

Building on the findings of this study, future research should prioritize longitudinal investigations to track the long-term impact of early mobilization (EM) on functional recovery, particularly in patients with prolonged ICU stays, to elucidate sustained benefits and optimize intervention timelines. Technology-enhanced mobilization strategies, such as telehealth-guided exercises or wearable sensors, could be piloted in understaffed units to bridge resource gaps and ensure consistent EM adherence. Furthermore, inclusive protocols must be developed for patients with pre-existing disabilities (e.g., paralysis), who were excluded here but represent a critical population requiring tailored rehabilitation approaches. Finally, economic analyses assessing the cost-benefit ratio of EM interventions in low-resource settings are urgently needed to advocate for funding, inform policy, and justify resource allocation. Together, these directions aim to address current gaps, enhance equity, and scale EM practices to improve outcomes in resource-constrained critical care environments.

## Conclusion

This study highlights the complex interplay of clinical, institutional, and patient-level factors influencing EM in Southern West Bank ICUs. While respiratory conditions, pain, and medical device dependency emerged as primary barriers, demographic factors played no significant role. The findings underscore the need for diagnosis-specific protocols,

interdisciplinary collaboration, and enhanced pain management to optimize EM practices. By addressing these challenges, healthcare providers in resource-limited settings can mitigate ICU-acquired complications, reduce hospitalization costs, and improve long-term patient outcomes. Future work should prioritize inclusive, longitudinal, and technology-driven approaches to bridge existing gaps in EM implementation.

## Supporting information

**S1 File. PERME ICU mobility tool.** The validated tool used to assess patient mobility levels and barriers in the ICU. (DOCX)

**S2 Text. Statistical analysis protocol.** Description of statistical methods, software, and sample size justification. (DOCX)

## Acknowledgments

The authors would like to express their thanks to the nurses who participated in the study.

## Author contributions

**Conceptualization:** Beesan Nassar, Moath Abu Ejheisheh, Ibrahim Aqtam.

**Data curation:** Beesan Nassar, Ahmad Batran.

**Formal analysis:** Ahmad Ayed, Ahmad Batran.

**Investigation:** Beesan Nassar, Ibrahim Aqtam, Ahmad Ayed.

**Methodology:** Beesan Nassar, Moath Abu Ejheisheh, Ahmad Ayed.

**Project administration:** Beesan Nassar.

**Supervision:** Moath Abu Ejheisheh, Ahmad Batran.

**Writing – original draft:** Beesan Nassar, Ahmad Ayed.

**Writing – review & editing:** Moath Abu Ejheisheh, Ibrahim Aqtam, Ahmad Batran.

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
