## [Decision Letter · Decision Letter 0]

PONE-D-25-04322Factors Affecting Early Mobilization among Critically Ill Patients in Southern West Bank HospitalsPLOS ONE

Dear Dr. aqtam,

Thank you for submitting your manuscript to PLOS ONE. After careful consideration, we feel that it has merit but does not fully meet PLOS ONE’s publication criteria as it currently stands. Therefore, we invite you to submit a revised version of the manuscript that addresses the points raised during the review process.

We look forward to receiving your revised manuscript.

Kind regards,

Amirmohammad Khalaji

Academic Editor

PLOS ONE

Journal Requirements:

3. Peer review at PLOS ONE is not double-blinded (https://journals.plos.org/plosone/s/editorial-and-peer-review-process). For this reason, authors should include in the revised manuscript all the information removed for blind review.

4. We suggest you thoroughly copyedit your manuscript for language usage, spelling, and grammar. If you do not know anyone who can help you do this, you may wish to consider employing a professional scientific editing service.

6. We note that your Data Availability Statement is currently as follows: All relevant data are within the manuscript and its Supporting Information files.

7. Please remove all personal information, ensure that the data shared are in accordance with participant consent, and re-upload a fully anonymized data set.

Reviewers' comments:

Reviewer's Responses to Questions

**Comments to the Author**

1. Is the manuscript technically sound, and do the data support the conclusions?

Reviewer #1: Partly

Reviewer #2: Yes

2. Has the statistical analysis been performed appropriately and rigorously? 

Reviewer #1: Yes

Reviewer #2: I Don't Know

3. Have the authors made all data underlying the findings in their manuscript fully available?

Reviewer #1: Yes

Reviewer #2: Yes

4. Is the manuscript presented in an intelligible fashion and written in standard English?

Reviewer #1: No

Reviewer #2: Yes

5. Review Comments to the Author

Reviewer #1: Dear authors,

thank you very much for the opportunity to review the submitted manuscript. The manuscript includes a study about barriers towards early mobilization of critically ill patients.

I am appreciating this work and have got some minor and serious concerns.

It is always a good idea to use line numbers, so that reviewers can easily refer to specific parts of the manuscript and avoid misunderstandings.

SERIOUS CONCERNS

Abstract: there are never “discussions” in section abstract, but “conclusions”. Please revise.

You have no reported results for “staff training, resource allocation, and other strategies”; hence, you cannot draw any conclusions about it. Add it to results or delete it from conclusions.

Page 4-Page 7: You have a part, called “literature review”. First, you report no method for this review, and hence, it is highly biased. Second, adding this “literature review” to section introduction makes the introduction extensively lengthy and wordy. Such a kind of literature review might be required in a master’s thesis, but not in a publication. You just report the current state of knowledge in your subject. My strong recommendation: move the part of “literature review” before the research question on page 4 (and delete the term “literature review”), and reduce the complete introduction to max 1.5 pages, but keep the references.

Population: I do not understand this. In section introduction you refer to critically ill patients, treated on Intensive Care Units. You included ICU patients with a stay of 2-5 days. Purpose of this study was “investigate the factors that influence EM among critically ill patients in XXX hospitals” (p. 4). Well done. But you recruited patients on medical wards, too, and there seems to be one rehab facility, too? Patients on medical wards and in rehab facilities are not critically ill, they are survivors of critical illness. This is a different population and a serious concern: what is your population?

Second, you excluded patients with “motor disabilities, communication disabilities,” (page 8), but according to table 3, 26% were unresponsive, table 4: 40% could not follow commands, 50% could not raise the leg against gravity. These are all conditions representing your exclusion criteria but were included. This is another serious concern.

Page 9 translation: for being a valid translation, please report details about the translation process. I hope that it has been performed using a scientific method for translation and cultural adaption.

MINOR CONCERNS

Page 2, Abstract, Methods: the number of 150 patients belong into results. Please report in- and exclusion criteria instead.

Page 2, Results: please report always both, number and percent

P8: since this study was observational, you do not need a required sample size. Please delete.

P10: please report the app you were using for the statistical analysis, such as R, SPSS, or else

Section results: please avoid redundant information in text and tables.

Section discussion: please reduce section discussion to your core findings and discuss in more depth, using varying arguments and literature.

Section limitations: please add much more limitations. Refer to www.equator-network.org, use an appropriate tool for your study, and add further limitation (what is missing).

Reviewer #2: The study presents a meaningful investigation into the barriers of early mobilization (EM) in critically ill patients, particularly in a resource-limited setting. It addresses an important gap in the literature by exploring regional factors in the Southern West Bank. The manuscript is generally well-organized, with a clear research question and appropriate methodology. However, it would benefit from greater clarity, improved precision in reporting, and more thoughtful contextualization of findings.

“Despite being well-documented, EM is usually not applied widely to critically ill patients, especially within resource-poor healthcare settings.”. Reference?

“The highest mean and SD of PERME ICU Mobility Scores were for abdominal pain patients at a mean of 28.0 (SD = 0.0)” : A standard deviation of 0.0 suggests only one patient or a data entry issue. Please justify this in the discussion.

“Pain management became a crucial barrier…” : This is one of the most important actionable findings. Consider recommending specific interventions (e.g., pre-activity analgesia, multidisciplinary pain teams).

The exclusion of patients with motor and communication disabilities…”: Consider explaining why this was necessary (e.g., due to limitations in the PERME score).

6. PLOS authors have the option to publish the peer review history of their article (what does this mean?). If published, this will include your full peer review and any attached files.

Reviewer #1: No

Reviewer #2: No

---

## [Author Response · Author response to Decision Letter 1]

12 Apr 2025

Dear Editor and Reviewers

We sincerely thank the Editor and Reviewers for their constructive feedback, which significantly strengthened the rigor and clarity of this work. Their expertise has been invaluable in contextualizing the study within the unique challenges of resource-limited ICUs.

Please to see attached Response for reviewers file

Dr Aqtam

---

## [Decision Letter · Decision Letter 1]

Factors Affecting Early Mobilization among Critically Ill Patients in Southern West Bank Hospitals

PONE-D-25-04322R1

Dear Dr. Aqtam,

We’re pleased to inform you that your manuscript has been judged scientifically suitable for publication and will be formally accepted for publication once it meets all outstanding technical requirements.

Kind regards,

Amirmohammad Khalaji

Academic Editor

PLOS ONE

Additional Editor Comments (optional):

Reviewers' comments:

Reviewer's Responses to Questions

**Comments to the Author**

1. If the authors have adequately addressed your comments raised in a previous round of review and you feel that this manuscript is now acceptable for publication, you may indicate that here to bypass the “Comments to the Author” section, enter your conflict of interest statement in the “Confidential to Editor” section, and submit your "Accept" recommendation.

Reviewer #1: All comments have been addressed

2. Is the manuscript technically sound, and do the data support the conclusions?

Reviewer #1: Yes

3. Has the statistical analysis been performed appropriately and rigorously? 

Reviewer #1: Yes

4. Have the authors made all data underlying the findings in their manuscript fully available?

Reviewer #1: Yes

5. Is the manuscript presented in an intelligible fashion and written in standard English?

Reviewer #1: Yes

6. Review Comments to the Author

Reviewer #1: Dear authors,

Thank you very much for the submission of a revised manuscript. You answered all concerns, or argued reasonable, and you improved your manuscript. There are no further concerns.

Thank you!

7. PLOS authors have the option to publish the peer review history of their article (what does this mean?). If published, this will include your full peer review and any attached files.

Reviewer #1: No

---

## [Editor Report · Acceptance letter]

PONE-D-25-04322R1

PLOS ONE

Dear Dr. Aqtam,

I'm pleased to inform you that your manuscript has been deemed suitable for publication in PLOS ONE. Congratulations! Your manuscript is now being handed over to our production team.

Kind regards,

on behalf of

Dr. Amirmohammad Khalaji

Academic Editor

PLOS ONE